# Exclusive Breastfeeding Rates at Hospital Discharge Across the Robson Ten-Group Classification System: A Retrospective Study

**DOI:** 10.3390/nu17233708

**Published:** 2025-11-26

**Authors:** Rafael Vila-Candel, Anna Martin-Arribas, Joaquín Mateu-Mollá, Fatima Leon-Larios, Desirée Mena-Tudela

**Affiliations:** 1La Ribera Health Department, Foundation for the Promotion of Health and Biomedical Research in the Valencian Region (FISABIO), 46020 Valencia, Spain; vila_rafcan@gva.es; 2Faculty of Health Sciences, Valencian International University (VIU), Pintor Sorolla 2, 46002 Valencia, Spain; joaquin.mateu@professor.universidadviu.com; 3School of Health Sciences Blanquerna, Universitat Ramon Llull, Carrer Padilla 326, 08025 Barcelona, Spain; 4Nursing Department, Faculty of Nursing, Physiotherapy and Podiatry, University of Seville, Avenzoar, 6, 41009 Seville, Spain; fatimaleon@us.es; 5Nursing Department, Faculty of Health Sciences, Feminist Institute, Universitat Jaume I, Avda. Sos I Baynat s/n, 12071 Castellón, Spain; dmena@uji.es

**Keywords:** caesarean section, robson classification, perinatal outcomes, exclusive breastfeeding, cross-sectional study

## Abstract

(1) Background: While the Robson Ten-Group Classification System (RTGCS) is widely used to assess and monitor caesarean section rates, its association with exclusive breastfeeding (EBF) outcomes at hospital discharge remains insufficiently explored. This study aimed to examine differences in EBF rates at hospital discharge across RTGCS groups among births attended at Hospital Universitario de la Ribera (Spain). (2) Methods: This retrospective observational study was conducted at a public hospital between 1 January 2010 and 31 December 2023. EBF at hospital discharge was analysed across Robson groups. Sociodemographic, obstetric, and neonatal variables were collected. A binomial logistic regression model was developed to identify predictors of EBF at discharge. Model fit was assessed using the Hosmer–Lemeshow goodness-of-fit test and Nagelkerke’s R^2^. (3) Results: The study analysed 23,081 births classified using the RTGCS, and 23,037 were included in the breastfeeding analysis. The overall EBF rate at discharge was 74.10%, with significant variation across Robson groups. Group 1 showed the highest EBF rate (78.339%) whereas Group 8 had the lowest (56.485%). Key factors positively associated with EBF included non-Spanish origin, nulliparity, cephalic presentation, singleton pregnancy, and term gestation. (4) Conclusions: Group 1 demonstrated the highest EBF rates, whereas Groups 8, 5, and 7 exhibited the lowest. These findings are essential for informing the development of targeted support strategies to improve breastfeeding outcomes in high-risk populations.

## 1. Introduction

Exclusive breastfeeding (EBF) is widely recognised as the optimal source of nutrition for newborns. Nevertheless, global EBF rates remain below the World Health Organization’s (WHO) 2025 target of 50% [1]. In Spain, only 39% of infants are exclusively breastfed at six months, according to national reports from the National Institute of Statistics [2]. Regional studies reveal considerable variability, with rates ranging from 81.5% at hospital discharge to 42.5% at six months [3]. These figures underscore the persistent challenges in maintaining breastfeeding beyond the immediate postpartum period.

Caesarean birth is one of the key factors that may hinder both the initiation and continuation of breastfeeding [4,5]. Although emergency caesarean sections are particularly associated with greater initial challenges in establishing breastfeeding, the broader influence of different obstetric conditions remains inconclusive [3,5]. Addressing these variations requires a more comprehensive framework capable of integrating multiple obstetric factors simultaneously—such as the Robson Ten-Group Classification System (RTGCS).

The RTGCS, developed in 2001, is an internationally recognised tool used to monitor and compare caesarean delivery rates across different settings [6,7]. The system classifies all women giving birth according to core obstetric characteristics, including parity, previous caesarean section, onset of labour, fetal presentation, and gestational age [6]. Given that the Robson groups are defined by variables known to influence breastfeeding initiation, it is plausible that this classification system may also help identify patterns in EBF outcomes [8,9,10,11,12].

However, no published studies have directly examined EBF rates at hospital discharge across the ten Robson groups. To address this gap, the primary aim of the present study was to determine whether EBF rates at hospital discharge differ across Robson groups, using data from 23,081 births attended at Hospital Universitario de la Ribera (HULR) between 2010 and 2023. The secondary objectives were to identify relevant sociodemographic and obstetric characteristics associated with EBF and to assess the independent impact of key predictors on breastfeeding outcomes using regression models.

## 2. Materials and Methods

### 2.1. Study Design, Population, and Sample

This study employed a retrospective observational design based on medical records, analysing all births (including both live births and stillbirths) that occurred at HULR between 1 January 2010 and 31 December 2023. HULR is a public hospital located in the eastern region of Spain, serving a population of approximately 250,000 inhabitants and attending an average of 1300 births annually. It is the only hospital in Spain with data publicly available on the RTGCS WHO platform.

Data were collected by reviewing the electronic medical records of all births during the study period; therefore, the study population and sample were effectively identical. Any missing data related to variables required for RTGCS classification were recovered through manual review of clinical records. Importantly, there were no missing data for the breastfeeding variable.

### 2.2. Analysis Criteria and Exclusions

General analysis (RTGCS): Both live births and stillbirths were included. This approach is consistent with World Health Organization (WHO) recommendations, which endorse the use of the RTGCS for analysing all births—regardless of neonatal outcome—to ensure a standardised and comprehensive evaluation of obstetric care.Analysis of Breastfeeding Rates:

Inclusion: The analytic subset was restricted exclusively to mother-live newborn dyads.

Exclusion: Stillbirths were explicitly excluded from this analytic subset to ensure that the evaluation focused on neonatal outcomes compatible with breastfeeding.

### 2.3. Data Collection

The RTGCS was used to classify caesarean sections within the study population. An overview of the criteria defining each group is presented below:-Group 1: Nulliparous, singleton cephalic, ≥37 weeks, spontaneous labour.-Group 2a: Nulliparous, singleton cephalic, ≥37 weeks, induced labour.-Group 2b: Nulliparous, singleton cephalic, ≥37 weeks, caesarean delivery before labour.-Group 3: Multiparous, singleton cephalic, ≥37 weeks, spontaneous labour.-Group 4a: Multiparous, singleton cephalic, ≥37 weeks, induced labour.-Group 4b: Multiparous, singleton cephalic, ≥37 weeks, caesarean delivery before labour.-Group 5: Previous caesarean delivery, singleton cephalic, ≥37 weeks, spontaneous or induced labour, or caesarean delivery before labour (VBAC).-Group 6: Nulliparous singleton breech, spontaneous or induced labour, or caesarean delivery before labour.-Group 7: Multiparous singleton breech (including previous caesarean delivery), spontaneous or induced labour, or caesarean delivery before labour.-Group 8: Multiple pregnancies, spontaneous or induced labour, or caesarean delivery before labour.-Group 9: Abnormal singleton lies (excluding breech but including prior caesarean delivery), spontaneous or induced labour, or caesarean delivery before labour.-Group 10: Singleton cephalic, ≤36 weeks (including previous caesarean delivery), spontaneous or induced labour, or caesarean delivery before labour.

An anonymised dataset of all births occurring within the study period was extracted from the hospital’s electronic medical records by the systems department. The dataset included all variables required for classification according to the RTGCS and was subsequently analysed by the research team. Classification was based on five key variables: parity and history of previous caesarean delivery; onset of labour (spontaneous, induced, or pre-labour caesarean); fetal presentation (cephalic, breech, or transverse); number of fetuses; and gestational age (preterm or full term).

The absolute contribution of a Robson group refers to the percentage of caesarean sections in that group relative to the total number of births (both vaginal and caesarean). This metric provides a population-level perspective, helping to identify the groups that most strongly influence the overall caesarean section rate. In contrast, the relative contribution represents the percentage of caesarean sections in a specific group relative to the total number of caesarean sections performed. This measure reflects each group’s proportional weight within the overall caesarean distribution and is essential for prioritising intervention areas with the greatest potential to reduce the overall rate. Both metrics are complementary and together support the optimisation of obstetric practice.

Additional variables collected included:

Sociodemographic characteristics: maternal age (in completed years) and country of origin (Spain vs. non-Spanish origin).

Obstetric and perinatal characteristics: neonatal sex (male/female); birth weight (grams); parity; umbilical cord arterial pH; Apgar scores at 1 and 5 min; fetal death (yes/no); admission to the Neonatal Intensive Care Unit (NICU) (yes/no); moderate prematurity (<32 weeks) (yes/no); and mode of birth (vaginal or caesarean). Early initiation of breastfeeding (EIBF) was categorised according to the time elapsed from birth to the first breast contact, distinguishing three key intervals: ≤30 min, 31–60 min, and ≥61–120 min.

Exclusive breastfeeding (EBF) was categorised as exclusive breastfeeding (Yes) versus mixed feeding or formula feeding (No). The average length of hospital stay in our centre is 48 h for vaginal births and 72 h for caesarean births. This variable was considered relevant, as it may influence EBF outcomes at discharge. Maternity nurses routinely recorded the type of feeding in the electronic medical record at the time of discharge, ensuring a standardised and reliable measure across all cases.

### 2.4. Statistical Analysis

A descriptive statistical analysis was conducted to examine the sociodemographic, obstetric, and neonatal characteristics of the study sample. Continuous variables were presented as means and standard deviations (SD), and categorical variables as frequencies and percentages.

To assess univariate associations, the chi-square (χ^2^) test was used. Odds ratios (OR) with 95% confidence intervals (CI) were calculated to evaluate the association between Robson groups and EBF at discharge. For the main EBF analysis, Robson subgroups 2a/2b and 4a/4b were combined into Group 2 and Group 4, respectively. The strength of association between categorical variables was measured using Pearson’s Phi (for 2 × 2 tables) or Cramer’s V (for larger contingency tables).

Binomial logistic regression analysis was then performed to assess the association between predictor variables and EBF at discharge. Model fit was evaluated using the Hosmer–Lemeshow goodness-of-fit test, and Nagelkerke’s R^2^ was calculated to estimate the proportion of variance explained.

For all analyses, a *p*-value < 0.05 was considered statistically significant. Statistical analyses were performed using SPSS, version 28.

## 3. Results

### 3.1. Descriptive Analysis

The sample consisted of 23,081 records classified according to the RTGCS. Sociodemographic, obstetric, and neonatal characteristics are presented in Table 1.

The overall EBF rate at discharge was 74.10%. Rates were 74.62% among vaginal births, 72.74% following emergency caesarean sections, and 69.28% after elective pre-labour caesarean sections, with statistically significant differences observed between these groups.

### 3.2. Bivariate Analysis of EBF and Perinatal Variables

Table 2 presents the distribution of EBF at discharge across the different RTGCS groups. The results indicate significant differences in the likelihood of EBF at discharge among the groups. Group 1 showed the highest probability of EBF, whereas Groups 8, 5, and 7 exhibited the lowest rates. EBF prevalence at discharge ranged from 56.49% in Group 8 to 78.34% in Group 1. Corresponding odds ratios (ORs) ranged from 0.359 (Group 8) to 0.973 (Group 2), with Group 2 being the closest to the reference category (Group 1).

To assess the temporal stability of EBF patterns, a comparative analysis was conducted using the observed frequencies for the Robson subgroups and for the total sample across two time periods: 2010–2016 and 2017–2023. An overall increase in EBF was observed in 2017–2023 compared with 2010–2016 (*p* = 0.017). When examining the individual Robson groups, increases in EBF were noted for Groups 1 and 2 (*p* < 0.001 for both), whereas decreases were observed in Groups 3 (*p* = 0.014) and 7 (*p* = 0.023). The remaining groups remained stable over time (*p* > 0.050) (Table 3). Subgroups 2a and 2b were combined into Group 2, as no significant differences in EBF rates were detected between them (77.875% vs. 75.909%; *p* = 0.498). Similarly, subgroups 4a and 4b were merged into Group 4 due to the absence of significant differences in EBF (71.189% vs. 68.421%; *p* = 0.708).

Additionally, we analysed the overall impact of the variables included in the RTGCS on EBF outcomes at hospital discharge, as well as other perinatal factors known to influence breastfeeding prevalence. Forty-four stillbirths were excluded from the analysis of breastfeeding rates, resulting in a final sample of 23,037 live births (Appendix A).

As shown in Table 4, variables such as mode of delivery, fetal presentation, and gestational age at birth were significantly associated with EBF prevalence, which reached 74.24% in the total sample. However, the strength of these associations was generally weak. Vaginal births were associated with slightly higher EBF rates compared with caesarean sections. Regarding fetal presentation, non-cephalic presentations (breech and oblique/transverse) were significantly associated with lower EBF rates at discharge compared with cephalic presentations. Term births were more strongly associated with EBF than preterm births. In addition, EBF rates at discharge were higher among newborns who were not admitted to the NICU than among those requiring NICU admission. A higher prevalence of EBF was also observed among neonates with Apgar scores > 7 at five minutes compared with those with scores ≤ 7, and among singleton births compared with multiple births. Expected percentages represent the frequencies that would be predicted for each cell in the contingency table if the variables were statistically independent (i.e., if no association existed). These values are calculated by multiplying the marginal row total by the marginal column total for each cell and dividing the result by the total sample size.

### 3.3. Caesarean Section Distribution by RTGCS

The overall caesarean section rate was 19.29%, with substantial variation across the ten Robson groups. Key differences in group distribution are detailed in Table 5. The greatest relative contribution to the total caesarean rate over the study period was observed in Group 2 (31.83%), followed by Group 1 (18.24%) and Group 6 (11.52%).

### 3.4. Multivariate Logistic Regression Analysis

A binomial logistic regression analysis was performed (Table 6), incorporating the variables encompassed by the RTGCS: parity and history of previous caesarean delivery; mode of labour onset (spontaneous, induced, or pre-labour caesarean); fetal presentation (cephalic, breech, or transverse); number of fetuses; and gestational age (preterm or full term). Additionally, variables that showed statistical significance in the bivariate analysis were included. We observed that non-Spanish maternal origin, nulliparity, cephalic presentation, singleton pregnancy, and gestational age ≥ 37 weeks were positively associated with EBF at discharge (*p* < 0.001 in all cases). Newborn sex (male) and maternal age were also positively associated, although with substantially smaller effect sizes (*p* < 0.050). Mode of labour onset was not associated with EBF at discharge (*p* = 0.905).

Women whose country of origin was not Spain (OR = 3.452), nulliparous women (OR = 1.394), those with cephalic fetal presentation (OR = 1.286), singleton pregnancies (OR = 2.148), and term gestations (OR = 1.378) were all significantly more likely to exclusively breastfeed at discharge (all *p* < 0.001, except fetal presentation *p* = 0.001). The model demonstrated good classification ability, correctly classifying 74.30% of cases; however, its modest explained variance (R^2^ = 0.065) indicates that additional unmeasured factors likely influence exclusive breastfeeding outcomes.

## 4. Discussion

The RTGCS is widely regarded as an effective method for monitoring and comparing caesarean section rates across different populations and obstetric practices worldwide [13]. The aim of this study was to examine whether differences existed in the prevalence of EBF at hospital discharge among the various groups defined by the RTGCS.

### 4.1. Variations in EBF Rates Across RTGCS Groups: Variations in EBF Rates Across RTGCS Groups

The study demonstrated significant variation in EBF at hospital discharge across the Robson groups. Overall, EBF prevalence was higher following vaginal birth (74.62%) compared with emergency caesarean sections (72.74%) and elective caesarean sections (69.28%), with statistically significant differences across modes of birth.

Reference Groups (Low Obstetric Risk): Group 1 (nulliparous, singleton, cephalic, term, spontaneous labour) served as the reference category and showed the highest likelihood of EBF at discharge (78.34%). Group 2 (nulliparous women with induced labour or pre-labour caesarean delivery) also exhibited a high EBF rate (77.86%), with an odds ratio close to that of Group 1 (OR = 0.973). This similarity may reflect the high initial motivation and strong self-efficacy often observed in nulliparous women, which can facilitate successful EBF at discharge [14].

However, the high caesarean rate observed in Group 2 (31.11%), together with its substantial relative contribution to the overall caesarean rate (31.83%), underscores the importance of reviewing induction protocols—even within this seemingly lower-risk group—to minimise potential negative effects on the birth experience and subsequent breastfeeding outcomes.

Evidence consistently suggests that multiparity is associated with more favourable breastfeeding outcomes [15]. However, in this study, the protective effect of multiparity was markedly attenuated—or even nullified—by the specific obstetric context defined by the Robson Classification, which appears to act as a strong mediating factor. Multiparous women in Groups 3 and 4 showed significantly lower EBF rates compared with nulliparous women in Group 1 (OR = 0.679 and OR = 0.689, respectively). These findings suggest that the complications or induction of labour characteristic of these groups may override the expected advantage conferred by previous breastfeeding experience [16,17]. This highlights the need for breastfeeding support strategies that are tailored not only to parity but also to the distinct risk profiles associated with each Robson group [18].

Groups with increased risk of EBF failure associated with caesarean section: Groups 5 (VBAC) and 7 (multiparous with breech presentation) recorded the lowest EBF rates after Group 8 (multiple gestations). The risk of EBF failure was significantly higher in Group 5 (OR = 0.492) and Group 7 (OR = 0.500). This negative association is largely explained by the exceptionally high caesarean section rates that characterise these groups (66.57% and 96.82%, respectively). Caesarean delivery may interfere with the early establishment of breastfeeding by delaying skin-to-skin contact and early initiation, and through postoperative factors such as pain, reduced maternal mobility, or the effects of anaesthesia and analgesic medication [19,20].

Similarly, although Group 6 (nulliparous women with breech presentation) had an exceptionally high caesarean section rate (98.09%), its EBF rate reached 72.66%. While lower than that of Group 1, the moderate risk of EBF failure (OR = 0.735) may indicate that the high initial motivation and antenatal preparation commonly observed among nulliparous women partly mitigated the negative effects of a planned caesarean section [14].

Groups with major neonatal risk (prematurity and multiple gestations): Groups 8 and 10 showed the most compromised EBF rates.

Group 8 (multiple pregnancies) had the lowest EBF rate in the entire study (56.49%; OR = 0.359). This finding is consistent with the intrinsic challenges of breastfeeding multiples and the higher frequency of NICU admissions observed in this group [21,22,23]. Group 10 (preterm births) also showed a strong negative association with EBF (OR = 0.617), in line with the reduced neuromuscular maturity of preterm newborns—which limits effective sucking ability [21,24,25]—and their higher likelihood of requiring NICU admission [21].

Temporal trends and implications for targeted breastfeeding strategies: The HULR achieved a key milestone in breastfeeding promotion by securing Phase 1D of the Baby-Friendly Hospital Initiative (BFHI) recommendations [26]. This institutional advancement is reflected in our findings: a comparative analysis of EBF rates at hospital discharge between the 2010–2016 and 2017–2023 periods showed an overall improvement in the later period. However, this progress was not evenly distributed across all groups. When trends were analysed using the Robson Classification, increases in EBF were concentrated in the low-risk groups—Groups 1 and 2 (both nulliparous)—whereas Groups 3 and 7 experienced decreases.

These findings suggest that the implementation of peripartum support strategies—such as early skin-to-skin contact, rooming-in, on-demand feeding, avoidance of unnecessary supplements or teats, and the continuous training and awareness of maternity staff—has been particularly effective in consolidating EBF among low-risk populations. Nevertheless, the stagnation or decline observed in higher-risk groups underscores the need to strengthen and tailor these interventions to ensure their effectiveness among the most vulnerable obstetric populations.

### 4.2. Multivariate Analysis of RTGCS Classification Variables and Exclusive Breastfeeding at Discharge

The binomial logistic regression analysis confirmed that several obstetric characteristics included in the Robson Classification were significantly associated with EBF at discharge.

Positive obstetric factors: Low-risk obstetric characteristics—including nulliparity (OR = 1.394), cephalic presentation (OR = 1.286), singleton pregnancy (OR = 2.148), and term gestation (>37 weeks, OR = 1.378)—were all associated with a significantly higher likelihood of EBF at discharge.

Sociocultural factor: The strongest positive association was observed among women whose country of origin was not Spain (OR = 3.452), consistent with findings from other studies conducted in Spain [26,27]. This result likely serves as a proxy indicator of the influence of pro-breastfeeding cultural norms, as well as the strong social and family support networks commonly present in non-Spanish communities [28,29,30,31,32,33].

Factors without association (Mode of labour onset and EIBF): In contrast to findings reported in some studies [34], the mode of labour onset (spontaneous vs. induced) was not significantly associated with EBF at discharge (*p* = 0.905). Early initiation of breastfeeding (EIBF) was excluded from the final model based on rigorous methodological criteria (low Wald statistics). This result is consistent with evidence indicating that EIBF is a stronger predictor of sustained EBF at later time points (e.g., 3 and 6 months) than of immediate success at hospital discharge (48–72 h) [20,26,35,36]. The model demonstrated good fit (Hosmer–Lemeshow test, *p* = 0.111) and correctly classified 74.30% of cases. However, its moderate explanatory power (Nagelkerke R^2^ = 0.065) suggests that the analysed obstetric and neonatal variables explain only a small proportion of the variance, an issue further explored in the limitations section.

The integration of results with the RTGCS is fundamental for identifying specific groups requiring targeted support. Our findings indicate that groups 5, 7, 8, and 10 exhibit significantly lower EBF rates. This highlights the need to focus intervention efforts on these high-risk populations, implementing tailored strategies, in line with other authors [24,30]. We acknowledge that EBF is a complex phenomenon and that our study analyses only a subset of determinants, underscoring that the combination of formal education and professional support is the most robust strategy for improving the initiation and continuation of breastfeeding [18,32].

### 4.3. Robson Classification

The World Health Organization (WHO) recommends a caesarean section rate of 10–15%. According to recent data, the average caesarean section rate in Spain stands at 25%, with notable differences between the public and private healthcare sectors, as well as across regions. In the public sector, the rate is 22.4%, whereas in the private sector it rises to 34.5%. Therefore, the rate observed in our study—19.289%—is below the national average and lower than those reported in both sectors [37].

Our findings show that most women included belonged to Group 1, followed by those in Group 3. A particularly noteworthy finding is that Group 2 presented the highest caesarean section rate (31.109%), exceeding the overall sample average. Although current evidence [34,38] suggests that labour induction in term, uncomplicated singleton pregnancies does not significantly increase the risk of caesarean delivery, our results indicate that nulliparity may be a key factor influencing this association. The higher incidence of caesarean sections observed in this group underscores the need to review induction protocols and more precisely evaluate the indications for labour induction, as highlighted in previous studies [10,39].

Our results confirm that caesarean section rates vary markedly across the different groups in the Robson Classification, with significantly elevated rates in Groups 5, 6, and 7—all of which considerably exceed the overall average and WHO recommendations. This pattern is consistent with previous evidence [8,9,10,40] indicating that Group 5 is the main contributor to the overall caesarean section rate in Europe, accounting for up to 95.0% of caesarean sections in Southern Europe. Furthermore, the predominance of caesarean births in Groups 6 and 7 suggests that current management protocols for these cases may influence the decision regarding the mode of delivery.

The observed differences may be attributed to multiple factors, including variations in clinical protocols, resource availability, and obstetric practices across healthcare centres. These findings highlight the need to review strategies aimed at reducing primary caesarean rates and to optimise decision-making in obstetric care, in alignment with international recommendations to improve maternal and neonatal outcomes

### 4.4. Strengths and Limitations

Our study presents several strengths that enhance the value and rigor of the findings. First, the use of the RTGCS as a standardised criterion for categorising births allows for objective comparison with other studies and facilitates the identification of patterns in caesarean section and breastfeeding rates [6]. Second, the large sample size provides a robust foundation for statistical analysis and enables the detection of significant differences between groups. This increases the reliability of the results and reduces the risk of biases typically associated with smaller samples [41,42].

However, our study also has several limitations. First, due to its observational design, it is not possible to fully control for confounding variables that may influence the outcomes. Additionally, the retrospective nature of the study means that the results depend heavily on the accuracy and completeness of the recorded data, which may introduce bias or result in missing variables. Furthermore, some relevant confounders may not have been documented, limiting the ability to adjust for them.

A key limitation is that EBF was assessed only at hospital discharge (48–72 h), an early time point that is susceptible to bias and does not allow evaluation of the persistence of EBF across Robson groups in the longer term. Moreover, the generalisability of our findings to other populations may be limited. Finally, although statistically significant differences were identified between groups, the effect sizes observed in most comparisons were weak [43,44]. This may be attributed to the large sample size, suggesting that the impact of the analysed variables on breastfeeding is likely small to moderate. Therefore, the results should be interpreted with caution in their clinical application, particularly given that many of the ORs are close to the threshold of statistical significance. Another relevant limitation is the explanatory capacity of the binary logistic regression model. Although it demonstrates a good overall fit and allows the extraction of clinically meaningful predictive dimensions, it explains only 6.5% of the variance in breastfeeding at discharge. This is expected, as the maintenance of EBF is determined by a complex interaction of factors that extend far beyond the clinical sphere, including psychosocial determinants (maternal intention, self-efficacy, and social or family support) and healthcare system factors (quality of care, BFHI compliance, etc.) [3,12,16,18,45,46]. Consequently, a substantial proportion of the variability in breastfeeding outcomes naturally lies beyond the explanatory scope of our clinical model, due to the methodological limitations inherent in capturing these external determinants [47].

Finally, as this is the first study to examine the relationship between EBF and the RTGCS, it provides an essential foundation for future research. Additional studies—particularly those incorporating prospective follow-up up to 6 months—are needed to determine whether the patterns observed at hospital discharge persist over time.

## 5. Conclusions

Our findings establish Group 1 as the benchmark for success while highlighting that Groups 8, 5, and 7 are the most affected, exhibiting the lowest EBF rates at discharge. Therefore, the use of the Robson Classification allows for the design of targeted, multidimensional interventions aimed at supporting and sustaining early breastfeeding success.

## Figures and Tables

**Table 1 nutrients-17-03708-t001:** Sociodemographic, obstetric, and neonatal characteristics of the study sample (N = 23,081).

			*n*		%
Maternal country of origin	Spain		18,019		78.069%
	Other than Spain		5062		21.931%
Neonatal sex	Male		11,819		51.207%
	Female		11,262		48.793%
Parity	0		12,731		55.158%
	>1		10,350		44.842%
Number of foetuses	Singleton		22,842		98.965%
	Multiple		239		1.035%
Previous caesarean section	No		22,476		97.379%
	Yes		605		2.621%
Onset of labour	Spontaneous		14,260		61.782%
	Induced		7602		32.936%
	Elective caesarean		1048		4.541%
	Emergency caesarean		171		0.741%
Mode of birth	Vaginal		18,629		80.711%
	Caesarean		4452		19.289%
Robson group	1		6851		29.682%
	2		4555		19.735%
	3		6437		27.889%
	4		2839		12.300%
	5		350		1.516%
	6		523		2.266%
	7		220		0.953%
	8		239		1.035%
	9		64		0.277%
	10		1003		4.346%
EBF at discharge	Yes		17,102		74.096%
	No		5979		25.904%
EIBF	30 min		1649		50.061%
	60 min		1384		42.016%
	120 min		261		7.923%
Stillbirth	No		23,037		99.809%
	Yes		44		0.191%
Moderate prematurity <32w	No		23,035		99.801%
	Yes		46		0.199%
Admission to NICU	No		20,484		88.748%
	Yes		2597		11.252%
	N	Minimum	Maximum	Mean	Std. Deviation
Maternal age	23,081	13	51	30.872	5.803
Umbilical cord arterial pH	21,264	0.000	7.450	7.240	0.339
Birth weight	21,884	250	5730	3.292.145	476.289
Gestational age at birth	23,081	24	43	39.191	1.558

EBF: Exclusive breastfeeding; EIBF: Early initiation of breastfeeding; NICU: Neonatal Intensive Care Unit.

**Table 2 nutrients-17-03708-t002:** Distribution of EBF at discharge by RTGCS in the study sample (N = 23,037).

Robson Group	*n*	Cs-Rate (%)	*χ* ^2^	OR (CI95%)	EBF at Discharge (%)	*χ* ^2^	OR (CI95%)
			*p*-Value	*p*-Value		*p*-Value	*p*-Value
1	6845	11.852%	5497.205 < 0.001	NA	78.339%	221.958 < 0.001	NA
2	4549	31.109%		3.358	77.863%		0.973
				(3.050–3.698)			(0.888–1.065)
				<0.001			0.465
3	6434	5.406%		0.425	71.064%		0.679
				(0.373–0.484)			(0.628–0.735)
				<0.001			<0.001
4	2834	15.604%		1.375	71.353%		0.689
				(1.213–1.558)			(0.623–0.761)
				<0.001			<0.001
5	349	66.571%		14.811	64.000%		0.492
				(11.722–18.713)			(0.392–0.616)
				<0.001			<0.001
6	520	98.088%		381.528	72.658%		0.735
				(203.182–716.418)			(0.601–0.898)
				<0.001			0.003
7	219	96.818%		226.303	64.384%		0.5
				(106.213–482.173)			(0.377–0.663)
				<0.001			<0.001
8	238	62.762%		12.535	56.485%		0.359
				(9.547–16.457)			(0.276–0.466)
				<0.001			<0.001
9	63	100.000%		0	71.875%		0.707
				0			(0.409–1.222)
				0.0966			0.214
10	986	25.823%		2.589	69.057%		0.617
				(2.208–3.036)			(0.533–0.715)
				<0.001			<0.001

NA: Not applicable; CS: caesarean section; *χ*^2^: Chi-squared test; OR: odds ratio; CI95%: Confidence Interval 95%, EBF: Exclusive breastfeeding. Stillbirths were excluded in the analysis of EBF at discharge. Note: Group 2 includes subgroups 2a and 2b; Group 4 includes subgroups 4a and 4b, according to the methodology for the exclusive breastfeeding (EBF) analysis.

**Table 3 nutrients-17-03708-t003:** Comparison of EBF between the time periods 2010–2016 and 2017–2023 for each Robson group (N = 23,037).

Robson Group		EBF at Discharge (%)		OR (CI95%)	*χ*^2^ (*p*-Value)
		2010–2016	2017–2023		
		N = 13,470	N = 9567		
Group 1	Observed	76.755%	80.545%	1.254 (1.114–1.411)	14.104 (<0.001)
	Expected	78.339%			
Group 2	Observed	75.889%	80.337%	1.298 (1.126–1.497)	12.886 (<0.001)
	Expected	77.863%			
Group 3	Observed	72.218%	69.381%	0.872 (0.782–0.972)	6.077 (0.014)
	Expected	71.064%			
Group 4	Observed	71.666%	70.849%	0.961 (0.813–1.136)	0.218 (0.640)
	Expected	71.353%			
Group 5	Observed	64.646%	63.158%	0.938 (0.604–1.456)	0.083 (0.774)
	Expected	64.000%			
Group 6	Observed	72.535%	72.803%	1.014 (0.689–1.491)	0.005 (0.945)
	Expected	72.658%			
Group 7	Observed	70.073%	54.878%	0.519 (0.294–0.917)	5.165 (0.023)
	Expected	64.384%			
Group 8	Observed	53.571%	60.606%	1.333 (0.791–2.248)	1.168 (0.280)
	Expected	56.485%			
Group 9	Observed	72.093%	71.429%	0.968 (0.304–3.080)	0.003 (0.956)
	Expected	71.875%			
Group 10	Observed	67.637%	71.173%	1.181 (0.894–1.561)	1.373 (0.241)
	Expected	69.057%			
Total	Observed	73.660%	75.050%	1.076 (1.013–1.142)	5.648 (0.017)
	Expected	74.237%			

*χ*^2^: Chi-squared test; OR: odds ratio; CI95%: Confidence Interval 95%; EBF: Exclusive breastfeeding. Stillbirths were excluded in the analysis of EBF at discharge.

**Table 4 nutrients-17-03708-t004:** Obstetric variables included in the RTGCS and perinatal outcomes associated with exclusive breastfeeding at hospital discharge. Forty-four stillbirths excluded. (N = 23,037).

Variable		*n*	EBF at Discharge (Observed%)	EBF at Discharge (Expected%)	*χ* ^2^		Phi or V	
					Value	*p*-Value	Value	*p*-Value
Mode of birth	Vaginal	18,597	74.783%	74.237%	15.018	<0.001	Phi = 0.026	<0.001
	Caesarean	4450	71.955%					
Foetal presentation	Cephalic	22,154	74.456%	74.237%	15.538	<0.001	V = 0.026	<0.001
	Breech	814	68.305%					
	Oblique-Transverse	69	73.913%					
Gestational age	>37	22,053	74.475%	74.237%	22.738	<0.001	V = 0.031	<0.001
	32–37	951	69.611%					
	<32	33	48.485%					
Multiple pregnancy	No	22,798	74.423%	74.237%	39.792	<0.001	Phi = 0.042	<0.001
	Yes	239	56.485%					
Apgar <7 at 5 min	No	22,921	74.338%	74.237%	24.205	<0.001	Phi = 0.032	<0.001
	Yes	116	54.310%					
NICU admission	No	20,440	74.560%	74.237%	9.866	0.002	Phi = 0.021	0.002
	Yes	2597	71.698%					

N: sample size; *χ*^2^: Chi-squared test; EBF: Exclusive Breastfeeding; NICU: Neonatal Intensive Care Unit; Pearson’s Phi and Cramer’s V are effect size measures (strength of association) for categorical variables, derived from the Chi-squared test. The interpretation of the values of Pearson’s Phi and Cramer’s V is as follows: 0.001–0.099 weak, 0.100–0.299 small, 0.300 to 0.499 moderate, and ≥ 0.500 large.

**Table 5 nutrients-17-03708-t005:** Distribution of caesarean births according to the RTGCS for births attended between 2010 and 2023 at the HULR (N = 23,081).

Group	Total Number of CS in Each Group	Total Number of Women Delivered in Each Group	Group Size	Group CS Rate	Absolute Group Contribution to Overall CS Rate	Relative Contribution of the Group to Overall CS Rate
1	812	6851	29.682%	11.852%	3.518%	18.239%
2	1417	4555	19.735%	31.109%	6.139%	31.828%
3	348	6437	27.889%	5.406%	1.508%	7.817%
4	443	2839	12.300%	15.604%	1.919%	9.951%
5	233	350	1.516%	66.571%	1010%	5.234%
6	513	523	2.266%	98.088%	2.223%	11.523%
7	213	220	0.953%	96.818%	0.923%	4.784%
8	150	239	1.035%	62.762%	0.650%	3.369%
9	64	64	0.277%	100.000%	0.277%	1.438%
10	259	1003	4.346%	25.823%	1.122%	5.818%
TOTAL	4452	23,081	100.000%	/	19.289%	100.000%

CS: Caesarean section. Note: Group 2 includes subgroups 2a and 2b; Group 4 includes subgroups 4a and 4b, according to the methodology for the exclusive breastfeeding (EBF) analysis.

**Table 6 nutrients-17-03708-t006:** Factors associated with exclusive breastfeeding at discharge: Binomial logistic regression analysis. Forty-four stillbirths excluded. (N = 23,037).

Summary of the Model		Hosmer and Lemeshow Test					
Log liKehood	R Nagelkerke	*χ* ^2^	df	*p*-Value			
25.243.561	0.065	13.023	8	0.111			
		**EBF**					
		Yes	No				
EBF	Yes	17,084	18	99.90%			
	No	5903	32	0.50%			
		**B**	**SE**	**Wald**	**df**	***p*-value**	**OR** **(CI95%)**
Age		0.008	0.003	7.720	1	0.005	1.008
							(1.002–1.013)
Country of origin	Other than Spain	1.239	0.049	650.053	1	<0.001	3.452
							(3.139–3.797)
Sex	Male	0.063	0.031	4.235	1	0.040	1.065
							(1.003–1.132)
Nulliparity	Yes	0.332	0.031	113.639	1	<0.001	1.394
							(1.312–1.482)
Previous caesarean	Yes	0.260	0.102	6.439	1	0.011	1.297
							(1.061–1.585)
Mode of labour onset	Spontaneous	0.004	0.032	0.014	1	0.905	1.004
							(0.942–1.069)
Presentation	Cephalic	0.252	0.078	10.377	1	0.001	1.286
							(1.104–1.499)
Number of foetuses	Singleton	0.765	0.137	31.354	1	<0.001	2.148
							(1.644–2.807)
Gestational age	>37 weeks	0.321	0.072	19.632	1	<0.001	1.378
							(1.196–1.588)

*χ*^2^: Chi-squared test; df: degrees of freedom; EBF: Exclusive breastfeeding; B: beta coefficient; SE: standard error. Note: Classification model: 74.300%.

## Data Availability

Data are available upon reasonable request. All necessary data are supplied and available in the manuscript; however, the corresponding author will provide the dataset upon request.

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
