# Peer review of "Exclusive Breastfeeding Rates at Hospital Discharge Across the Robson Ten-Group Classification System: A Retrospective Study"

_nutrients, 2025, doi:10.3390/nu17233708_

Round 1
Reviewer 1 Report
Comments and Suggestions for Authors
This is a well-written and timely retrospective study that investigates an under-explored application of the Robson Ten-Group Classification System (RTGCS). The use of a large, long-term dataset from a single center is a strength. The manuscript is generally clear, the methodology is sound for a retrospective design, and the statistical analysis is appropriate. The finding that the RTGCS can help identify groups of women at risk for not exclusively breastfeeding (EBF) at discharge is valuable for targeting clinical support. However, several major and minor points should be addressed to strengthen the manuscript before it is considered for publication.
Major Comments:
- Low Explained Variance in the Regression Model: The binomial logistic regression model, while statistically significant, explains only 6.5% of the variance in EBF outcomes (Nagelkerke's R²). The authors correctly acknowledge this limitation, but the discussion should be expanded. Please elaborate more specifically on what other critical factors, not captured in the clinical dataset, likely account for the remaining variance. These could include maternal education, socioeconomic status, prenatal intention to breastfeed, quality and timing of in-hospital lactation support, maternal body mass index, and social support at home. Contextualizing the model's modest explanatory power is crucial for a correct interpretation of the results.
- Rationale for Combining Robson Subgroups: The authors combined subgroups 2a/2b and 4a/4b for the primary EBF analysis, stating it aligns with standard practice for breastfeeding outcomes. While this simplifies the analysis, it potentially masks important clinical differences. For instance, the experience and potential challenges for a woman in Group 2a (induced labor) are likely very different from those in Group 2b (elective pre-labor cesarean). A stronger justification for this grouping is needed, perhaps supported by a preliminary analysis showing no significant EBF difference between the subgroups. Alternatively, a sensitivity analysis presenting results for the separate subgroups would be highly informative.
- Definition and Measurement of the EBF Outcome: The outcome variable "Exclusive breastfeeding at discharge" is central to the study. The manuscript states it was routinely recorded in the electronic medical record, but the method of assessment is not described. Was this based on maternal self-report, nurse observation, a standardized tool, or a combination? Clarifying the operational definition and validation of this key outcome measure is essential for assessing the validity and potential for misclassification bias.
- Generalizability and Single-Center Context: The study is conducted in a single public hospital in Spain with a cesarean section rate (19.3%) notably below the national average. The authors should more explicitly discuss how this specific context might limit the generalizability of the findings. For example, would the same EBF disparities across Robson groups be observed in a setting with a higher baseline CS rate or different cultural and institutional practices around breastfeeding support?
- Temporal Changes Over the Long Study Period: The data spans 14 years (2010-2023). During this time, clinical guidelines, breastfeeding promotion initiatives, and cesarean section practices likely evolved. The authors should address whether these temporal trends were considered. A sensitivity analysis, such as comparing EBF rates across Robson groups in two or three distinct time periods, could strengthen the findings by demonstrating that the observed associations are stable over time.
Minor Comments:
- Clarity of Results Tables: Table 2 is quite dense and challenging to read. Consider restructuring it for better clarity, for example, by separating the cesarean section metrics from the EBF metrics or by using a different layout to improve readability.
- Interpretation of "Non-Spanish Origin": The strong positive association between "non-Spanish origin" and EBF (OR=3.452) is a notable finding. The authors should briefly discuss potential reasons for this, such as cultural norms, stronger community support, or different socioeconomic factors among immigrant populations, while acknowledging the limitation of this broad categorization.
- Underutilized Data on Early Initiation: Table 1 includes data on the timing of early breastfeeding initiation (30, 60, 120 min), but this variable is not integrated into the main analysis or discussion. Given that early initiation is a known key factor for EBF success, especially after cesarean section, briefly discussing its distribution across Robson groups or including it in the regression model could add depth to the analysis.
- Statistical Terminology: In Table 3, the use of "Phi or V" is statistically correct but may not be familiar to all readers. Adding a brief footnote explaining that these are measures of effect size for categorical data would be helpful.
This manuscript presents a valuable analysis that extends the utility of the Robson Classification beyond cesarean section monitoring to the field of breastfeeding support. The authors have identified meaningful disparities in EBF rates across Robson groups. Addressing the major comments, particularly concerning the model's explained variance, the handling of subgroups, and the definition of the EBF outcome, will strengthen the manuscript and make it a candidate for publication.
Author Response
This is a well-written and timely retrospective study that investigates an under-explored application of the Robson Ten-Group Classification System (RTGCS). The use of a large, long-term dataset from a single center is a strength. The manuscript is generally clear, the methodology is sound for a retrospective design, and the statistical analysis is appropriate. The finding that the RTGCS can help identify groups of women at risk for not exclusively breastfeeding (EBF) at discharge is valuable for targeting clinical support. However, several major and minor points should be addressed to strengthen the manuscript before it is considered for publication.
Thank you for your very positive and constructive feedback on our manuscript.
We have considered all your comments and suggestions and the comments made by the other reviewers attempting to improve/refine the original manuscript.
Below, you will find a point-by-point response to your comments (in red).
Dear author’s
I was pleased to review your article and i have the following comments:
Major Comments:
Point 1: Low Explained Variance in the Regression Model: The binomial logistic regression model, while statistically significant, explains only 6.5% of the variance in EBF outcomes (Nagelkerke's R²). The authors correctly acknowledge this limitation, but the discussion should be expanded. Please elaborate more specifically on what other critical factors, not captured in the clinical dataset, likely account for the remaining variance. These could include maternal education, socioeconomic status, prenatal intention to breastfeed, quality and timing of in-hospital lactation support, maternal body mass index, and social support at home. Contextualizing the model's modest explanatory power is crucial for a correct interpretation of the results.
Response 1: Thank you for your comment. Indeed, a low value is obtained for Nagelkerke's R, whose function is to determine the percentage of the explained variance based on the predictor variables included in the model. However, from the perspective that the phenomenon under study involves enormous complexity, it can be interpreted as more acceptable. Your suggestion to expand the discussion on this matter is absolutely appropriate, so we have delved deeper into this issue to better contextualize it in the corresponding section, in page 15, lines 471-481 with news references supported.
Point 2: Rationale for Combining Robson Subgroups: The authors combined subgroups 2a/2b and 4a/4b for the primary EBF analysis, stating it aligns with standard practice for breastfeeding outcomes. While this simplifies the analysis, it potentially masks important clinical differences. For instance, the experience and potential challenges for a woman in Group 2a (induced labor) are likely very different from those in Group 2b (elective pre-labor cesarean). A stronger justification for this grouping is needed, perhaps supported by a preliminary analysis showing no significant EBF difference between the subgroups. Alternatively, a sensitivity analysis presenting results for the separate subgroups would be highly informative.
Response 2: We appreciate your insightful feedback on our study. A comparative analysis was conducted between Groups 2a and 2b to determine differences between them in EBF, as well as between Groups 4a and 4b. It is observed that both are homogeneous with respect to the variable. This information is included in text format, in result section, page 6, lines 192-196.
Point 3: Definition and Measurement of the EBF Outcome: The outcome variable "Exclusive breastfeeding at discharge" is central to the study. The manuscript states it was routinely recorded in the electronic medical record, but the method of assessment is not described. Was this based on maternal self-report, nurse observation, a standardized tool, or a combination? Clarifying the operational definition and validation of this key outcome measure is essential for assessing the validity and potential for misclassification bias.
Response 3: Thank you for your input. In order to clarify this sentence, we have rewritten the paragraph in page 3, lines 143-145.
Point 4: Generalizability and Single-Center Context: The study is conducted in a single public hospital in Spain with a cesarean section rate (19.3%) notably below the national average. The authors should more explicitly discuss how this specific context might limit the generalizability of the findings. For example, would the same EBF disparities across Robson groups be observed in a setting with a higher baseline CS rate or different cultural and institutional practices around breastfeeding support?
Response 4: We appreciate your comment regarding the limitations on generalisability stemming from the single-centre context and our lower caesarean section rate (19.3%). We acknowledge the validity of this point, which is already addressed in the Discussion section (page 14, lines 457-458), and we justify the selection of the HULR in the Methodology (page 2, lines 76-78) as it was the only hospital in Spain with RTGCS data publicly registered on the WHO platform, a key methodological requirement. Despite this limitation, our primary goal is not immediate generalisation but to establish the first baseline data on the relationship between EBF and the RTGCS in our country. The value of our work lies in the disparities observed (progress in Groups 1 and 2 versus decline in Groups 7, 8 and 10), which raise essential questions about how breastfeeding support interventions should be adapted. Therefore, while generalisability is limited, our findings serve as a critical catalyst for future research in settings with different caesarean section profiles and varying cultural practices. We have added a sentence in limitations section to clarify this relevant aspect in page 15, lines 458-461.
Point 5: Temporal Changes Over the Long Study Period: The data spans 14 years (2010-2023). During this time, clinical guidelines, breastfeeding promotion initiatives, and cesarean section practices likely evolved. The authors should address whether these temporal trends were considered. A sensitivity analysis, such as comparing EBF rates across Robson groups in two or three distinct time periods, could strengthen the findings by demonstrating that the observed associations are stable over time.
Response 5: Thank you for your relevant feedback. We agree on the importance of considering the evolution of clinical guidelines and institutional practices over the 14-year period A comparative analysis for EBF rates was conducted by dividing the time into two more discrete periods (2010–2016 and 2017–2023). An increase in total EBF is observed, as well as more specific differences between groups, which are detailed in both the table and the text.
We have taken the liberty of addressing this Major Point 5 alongside Minor Point 3 (on the early initiation of breastfeeding), as the stability of this variable is intrinsically relevant to both issues. The temporal analysis revealed an overall increase in EBF rates but with specific patterns across the Robson groups (increase in Groups 1 and 2, decrease in Groups 3 and 7), confirming that the observed associations are not uniformly stable. This information, along with the accompanying new Table 3, is detailed in the Results section (page 6, lines 187–193) and its integrated interpretation is addressed in the Discussion (page 12, lines 341-355).
Minor Comments:
Point 1: Clarity of Results Tables: Table 2 is quite dense and challenging to read. Consider restructuring it for better clarity, for example, by separating the cesarean section metrics from the EBF metrics or by using a different layout to improve readability.
Response 1: We sincerely appreciate your comment regarding the presentation of Table 2. We understand that its density may pose a challenge. However, we believe the table's current structure remains the most effective way to comprehensively capture and visualize in a single view the direct contrast between the Robson classification groups and the breastfeeding rates at discharge, which is crucial for our analysis. To mitigate the condensation problem without losing this comparative view, we propose to the journal that the table be included in a landscape (horizontal) orientation, which will allow for better visualization and readability of the data.
Point 2: Interpretation of "Non-Spanish Origin": The strong positive association between "non-Spanish origin" and EBF (OR=3.452) is a notable finding. The authors should briefly discuss potential reasons for this, such as cultural norms, stronger community support, or different socioeconomic factors among immigrant populations, while acknowledging the limitation of this broad categorization.
Response 2: Thank you for your input. We have rewritten this comment in Discussion section, page 13, lines 407-410, and in Limitations section page 15, lines 484-489.
Point 3: Underutilized Data on Early Initiation: Table 1 includes data on the timing of early breastfeeding initiation (30, 60, 120 min), but this variable is not integrated into the main analysis or discussion. Given that early initiation is a known key factor for EBF success, especially after cesarean section, briefly discussing its distribution across Robson groups or including it in the regression model could add depth to the analysis.
Response 3: Thank you very much for your interesting and insightful observation. The variable early initiation of breastfeeding was originally incorporated into the binary logistic regression model as a potential predictor variable. However, it was ultimately not selected for the final model composition after a succession of steps due to obtaining Wald scores below 1.000 and/or p-values exceeding 0.050. Nevertheless, it is considered a valuable variable, and further details regarding it have been expanded upon in the Discussion section (page 14, lines 421-428).
Point 4: Statistical Terminology: In Table 3, the use of "Phi or V" is statistically correct but may not be familiar to all readers. Adding a brief footnote explaining that these are measures of effect size for categorical data would be helpful.
Response 4: Thank you very much for the suggestion. The clarification is incorporated into the corresponding table footnote, page 9, lines 227-228.
This manuscript presents a valuable analysis that extends the utility of the Robson Classification beyond cesarean section monitoring to the field of breastfeeding support. The authors have identified meaningful disparities in EBF rates across Robson groups. Addressing the major comments, particularly concerning the model's explained variance, the handling of subgroups, and the definition of the EBF outcome, will strengthen the manuscript and make it a candidate for publication.
Thank you for your support, relevant and constructive comments.
Reviewer 2 Report
Comments and Suggestions for Authors
Introduction
The passage is too dense, break it into shorter paragraphs.
Phrases like “Robson classification” and “Robson Ten-Group Classification System (RTGCS)” are repeated multiple times. After first mention, use only the acronym.
Phrases like “Robson classification” and “Robson Ten-Group Classification System (RTGCS)” are repeated multiple times. After first mention, use only the acronym.
Methods
Many sentences are long and use multiple dependent clauses (e.g., “Data were collected by reviewing the electronic medical records of all births that occurred during the study period, thereby making the study population and sample effectively the same.”)
It’s not entirely clear which data were analyzed for breastfeeding (e.g., only live births? term infants?). Clarify your analytic subset.
Consider explicitly stating how missing data were handled.
Inclusion/exclusion not clearly separated.
Results
The results are comprehensive and data-rich, but the presentation is dense, fragmented, and occasionally unclear in interpretation.
The text contains broken lines, misplaced table fragments, and line-number remnants.
Divide results into distinct subsections such as: Descriptive Statistics, Bivariate Analysis of EBF and Perinatal Variables, Caesarean Section Distribution by Robson Group, Multivariate Logistic Regression.
Ensure consistent p-value notation — use either “p < 0.001” or “p = 0.04,” not a mix.
Author Response
Response to Reviewer 2 Comments
Thank you for your very positive and constructive feedback on our manuscript.
We have considered all your comments and suggestions (and the comments made by the other reviewer), attempting to improve/refine the original manuscript.
Below you will find a point-by-point response to your comments (in red).
Introduction.
Point 1: The passage is too dense, break it into shorter paragraphs.
Response 1: Thank you for your highlight. Amended in page 1, lines 44-69.
Point 2: Phrases like “Robson classification” and “Robson Ten-Group Classification System (RTGCS)” are repeated multiple times. After first mention, use only the acronym.
Response 2: Thank you for your comment. Amended. We have revised all of them and put the acronym.
Methods
Point 3: Many sentences are long and use multiple dependent clauses (e.g., “Data were collected by reviewing the electronic medical records of all births that occurred during the study period, thereby making the study population and sample effectively the same.”)
Response 3: Thank you for your comment. We have rewritten the paragraph as follows in page 2, lines 79-81.
Point 4: It’s not entirely clear which data were analyzed for breastfeeding (e.g., only live births? term infants?). Clarify your analytic subset.
Response 4: Thank you for your input. We have rewritten the paragraph as follows in page 2, lines 84-92.
Point 5: Consider explicitly stating how missing data were handled.
Response 5: Thank you for your comment. The missing data affecting the variables necessary to perform the RTGCS were obtained after manually reviewing each clinical record. We had no missing data regarding the type of breastfeeding. We have added this sentence in page 2, lines 81-83.
Point 6: Inclusion/exclusion not clearly separated.
Response 6: Thank you for your comment. Amended in section 2.2 Analysis criteria and Exclusions in page 2, lines 84-92.
Results
Point 7: The results are comprehensive and data-rich, but the presentation is dense, fragmented, and occasionally unclear in interpretation.
Response 7: Thank you for this suggestion. In order to help you more specifically, it would be useful to know: Which of your tables or figures do you think is the most dense or confusing, or which part of the Results section has received the most comments regarding? We acknowledge that some tables (previous Table 3) may appear fragmented or overly simple in the current manuscript. This is due to the layout and formatting restrictions of the submission manuscript. We trust that the final typesetting by the journal (post-acceptance) will allow for a more appropriate, complete, and less fragmented presentation of the tabulated data. To mitigate the condensation problem without losing this comparative view, we propose to the journal that the table be included in a landscape (horizontal) orientation, which will allow for better visualization and readability of the data.
Point 8: The text contains broken lines, misplaced table fragments, and line-number remnants.
Response 8: Thank you for this comment. Please, could you tell us the line numbers for these errors? We believe that this is due to the layout and formatting restrictions of the submission manuscript.
Point 9: Divide results into distinct subsections such as: Descriptive Statistics, Bivariate Analysis of EBF and Perinatal Variables, Caesarean Section Distribution by Robson Group, Multivariate Logistic Regression.
Response 9: Thank you for this suggestion. Amended in subsections, page 4 (line 164), page 5 (line 174), page 9 (line 229 and 240).
Point 10: Ensure consistent p-value notation — use either “p < 0.001” or “p = 0.04,” not a mix.
Response 10: The p-value of 0.04 in the corresponding table has been corrected to align with the general format of the text (p = 0.040), where three decimal places are provided in all cases. This appears to have been a formatting error during the word processor's value encoding. Thank you very much for your guidance.

Reviewer 3 Report
Comments and Suggestions for Authors
The topic is interesting because it brings an element of novelty by relating the breastfeeding rate to Robson's classes for cesareans.
Abstract
The abstract is well-structured, following the typical sections (Background, Methods, Results, Conclusions). It is fairly clear and provides a comprehensive overview of the study, including important details such as the number of participants, variables analyzed, and main findings. However, some sentences are somewhat long and complex. Shortening certain sentences could enhance readability. For the results, summarizing the clinical or practical significance of differences between Robson groups—beyond percentages—would be beneficial. The conclusions are too long for an abstract.
- Introduction
The introduction fulfills its purpose by emphasizing the importance of breastfeeding and the associated challenges, such as low rates of exclusive breastfeeding and the impact of cesarean delivery. The role of the Robson classification has been clearly outlined, and the research gap that your study aims to address is explicitly specified. Additionally, the primary and secondary objectives of your study are clearly articulated.
However, some modifications could improve the text.
You should include the sample size in this paragraph.
"Stillbirths were excluded in the analysis of breastfeeding rates."
The inclusion and exclusion criteria are not specified.
Does the 39% figure from the National Health Survey refer specifically to exclusive breastfeeding?
Evidence regarding differences between Robson classification groups and the initiation of breastfeeding is limited. What evidence? References should be inserted.
Some sentences are lengthy and could be simplified to improve the flow.
More fluid transitions between paragraphs would make the text more cohesive, as it currently feels somewhat disjointed.
While existing evidence has been described, it should be synthesized more concisely to better highlight the research gap your study addresses.
- Materials and Methods
Some parts are very detailed, but could be made more concise without losing essential information. For example, the explanation of absolute and relative contribution metrics could be summarized in a more straightforward manner.
Some sentences are very long and could be broken down into shorter, clearer segments. In particular, variable descriptions or statistical analyses can be made more concise.
Regarding the inclusion of stillbirths:
"Stillbirths were included and analysed in the study to ensure a comprehensive assessment of all pregnancy outcomes within the hospital setting. Their inclusion is consistent with the recommendations of the World Health Organization (WHO), which endorses the use of the RTGCS for analysing all births—regardless of neonatal outcome—to facilitate a standardized and inclusive evaluation of obstetric care."
It should be clarified how these cases were included if the focus is on breastfeeding, which occurs after birth. Was their inclusion considered only in relation to data not involving breastfeeding?
This point needs clarification, as including stillbirths could introduce bias if not properly justified.
- Results
This section presents detailed results regarding the impact of specific variables (such as mode of delivery, fetal presentation, gestational age, etc.) on exclusive breastfeeding at discharge.
First, descriptive data are provided (e.g., overall EBF percentage, differences between groups like vaginal vs. cesarean, fetal presentations, etc.).
Then, a logistic regression analysis identifies which variables are significantly associated with EBF, including relevant factors such as country of origin. The model shows good predictive capacity, with a moderate R², indicating that other unmeasured factors may influence EBF.
Would it be helpful to organize the results into clear subsections, such as: Descriptive Results, Bivariate Analysis, Multivariate Analysis?
Avoid excessive repetition and overly long or complex sentences to improve clarity.
- Discussion
The discussion provides a solid analysis of the study findings, addressing aspects such as differences in exclusive breastfeeding rates across Robson groups, associations with perinatal outcomes, and obstetric practice implications. It also compares results with existing literature, which is positive, and acknowledges methodological limitations, such as the observational design and potential biases inherent in retrospective data.
However, some points could be improved:
- Long and complex sections should be broken into shorter paragraphs for better readability.
- Emphasize the main results and practical implications more clearly, avoiding excessive technical details or numerical data that may distract the reader.
- The sentence:
"This finding underscores that motivation alone may be insufficient and highlights the need for enhanced professional support during hospitalization to capitalize on this initial drive and mitigate the potential adverse effects of the obstetric intervention [24]."
The study does not prove it. Should be clarified, as it may imply causality not demonstrated by the study.
- The statement:
"This suggests that multiparity does not always guarantee higher EBF rates, possibly influenced by prior unsatisfactory experiences."
can be expanded to mention other influencing factors.
- The claim:
"These findings are consistent with some previous studies [26,27]."
should be balanced by citing studies with conflicting results.
- Regarding breech presentation and cesarean delivery:
"Breech presentations that result in caesarean births may delay the initiation of breastfeeding and impair the mother’s ability to breastfeed [34–36]."
How does breech presentation delay the start of breastfeeding? It would be helpful to specify how breech presentation delays breastfeeding initiation.
- The sentence:
"Although higher Apgar scores facilitated effective breastfeeding at 36 hours, this study did not find a clear association with long-term success [44]." Which study? If it is yours, the breastfeeding data have been analyzed until discharge and not in the long term, so this sentence seems improper. Should specify which study is referenced. If your data only cover discharge, long-term conclusions are limited.
Speaking of this, among the limitations in my opinion is the fact that the data are related to discharge and therefore, as interesting as they are, they are a bit limiting because I imagine that you too discharge early at 48-72 hours maximum and it should be noted that the most significant data would be that relating to the following months. The limitations related to data collection at discharge—often within 48-72 hours—and the absence of follow-up data over months should be explicitly acknowledged.
- Also, discuss how these limitations influence interpretation and future research.
- Conclusions
Conclusions should be more concise and impactful, clearly summarizing the key messages without repetition.
Author Response
Thank you for your very positive and constructive feedback on our manuscript.
We have considered all your comments and suggestions (and the comments made by the other reviewer), attempting to improve/refine the original manuscript.
Below you will find a point-by-point response to your comments (in red).
The topic is interesting because it brings an element of novelty by relating the breastfeeding rate to Robson's classes for cesareans.
Abstract
The abstract is well-structured, following the typical sections (Background, Methods, Results, Conclusions). It is fairly clear and provides a comprehensive overview of the study, including important details such as the number of participants, variables analyzed, and main findings. However, some sentences are somewhat long and complex. Shortening certain sentences could enhance readability. For the results, summarizing the clinical or practical significance of differences between Robson groups—beyond percentages—would be beneficial. The conclusions are too long for an abstract.
- Introduction
The introduction fulfills its purpose by emphasizing the importance of breastfeeding and the associated challenges, such as low rates of exclusive breastfeeding and the impact of cesarean delivery. The role of the Robson classification has been clearly outlined, and the research gap that your study aims to address is explicitly specified. Additionally, the primary and secondary objectives of your study are clearly articulated.
However, some modifications could improve the text.
Point 1: You should include the sample size in this paragraph.
Response 1: Thank you for your highlight. Amended in page 2, line 66.
Point 2: "Stillbirths were excluded in the analysis of breastfeeding rates."
Response 2: Thank you for your comment. In order to clarify the exclusion criteria, we have rewritten a 2.2 item in page 2, lines 84-92.
Point 3: The inclusion and exclusion criteria are not specified.
Response 3: Thank you for your comment. Amended in response 2.
Point 4: Does the 39% figure from the National Health Survey refer specifically to exclusive breastfeeding?
Response 4: Thank you for your input. We confirm that in our context, data from the National Institute of Statistics through the National Health Survey indicate that only 39% of babies were under EBF at six months [2]. Page 2, lines 46-47.
Point 5: Evidence regarding differences between Robson classification groups and the initiation of breastfeeding is limited. What evidence? References should be inserted.
Response 5: Thank you for your comment. We have rewritten the sentence in page 2, lines 63-64.
Point 6: Some sentences are lengthy and could be simplified to improve the flow.
Response 6: Thank you for your comment. Amended in page 2.
Point 7: More fluid transitions between paragraphs would make the text more cohesive, as it currently feels somewhat disjointed.
Response 7: Thank you for this constructive suggestion. To address this, and in line with another comment from Reviewer 2, we have restructured the Introduction section. Crucially, we have also carefully revised the opening and closing sentences of each paragraph to ensure the transitions are now more fluid and logical, thereby creating a significantly more cohesive narrative throughout the text.
Point 8: While existing evidence has been described, it should be synthesized more concisely to better highlight the research gap your study addresses.
Response 8: Thank you for this suggestion and amended (response 7).
Methods
Some parts are very detailed, but could be made more concise without losing essential information. For example, the explanation of absolute and relative contribution metrics could be summarized in a more straightforward manner.
Point 9: Some sentences are very long and could be broken down into shorter, clearer segments. In particular, variable descriptions or statistical analyses can be made more concise.
Response 9: Thank you for this suggestion. We have thoroughly revised the section to address this comment, ensuring a more concise synthesis of the background evidence, in page 3 lines 123-131, and page 4, lines 147-162.
Point 10: Regarding the inclusion of stillbirths:
"Stillbirths were included and analysed in the study to ensure a comprehensive assessment of all pregnancy outcomes within the hospital setting. Their inclusion is consistent with the recommendations of the World Health Organization (WHO), which endorses the use of the RTGCS for analysing all births—regardless of neonatal outcome—to facilitate a standardized and inclusive evaluation of obstetric care."
It should be clarified how these cases were included if the focus is on breastfeeding, which occurs after birth. Was their inclusion considered only in relation to data not involving breastfeeding?
This point needs clarification, as including stillbirths could introduce bias if not properly justified.
Response 10: Thank you for your comment. In order to clarify this sentence, we have rewritten the paragraph as follows in page 2, line 73, and 84-92.
Results
This section presents detailed results regarding the impact of specific variables (such as mode of delivery, fetal presentation, gestational age, etc.) on exclusive breastfeeding at discharge.
First, descriptive data are provided (e.g., overall EBF percentage, differences between groups like vaginal vs. cesarean, fetal presentations, etc.).
Then, a logistic regression analysis identifies which variables are significantly associated with EBF, including relevant factors such as country of origin. The model shows good predictive capacity, with a moderate R², indicating that other unmeasured factors may influence EBF.
Point 11: Would it be helpful to organize the results into clear subsections, such as: Descriptive Results, Bivariate Analysis, Multivariate Analysis?
Response 11: Thank you for your suggestion. Amended in different subsections in page 4 (line 164), page 5 (line 174), page 9 (line 229 and 240)
Point 12: Avoid excessive repetition and overly long or complex sentences to improve clarity.
Response 12: Thank you for your comment. We have revised and adapted some sentences in page 8 lines 219-223.
Discussion
The discussion provides a solid analysis of the study findings, addressing aspects such as differences in exclusive breastfeeding rates across Robson groups, associations with perinatal outcomes, and obstetric practice implications. It also compares results with existing literature, which is positive, and acknowledges methodological limitations, such as the observational design and potential biases inherent in retrospective data.
However, some points could be improved:
Point 13: Long and complex sections should be broken into shorter paragraphs for better readability.
Response 13: Thank you for your comment. Amended in different subsections
Point 14: Emphasize the main results and practical implications more clearly, avoiding excessive technical details or numerical data that may distract the reader.
Response 14: Thank you for your comment. Amended in different subsections
Point 15: The sentence:
"This finding underscores that motivation alone may be insufficient and highlights the need for enhanced professional support during hospitalization to capitalize on this initial drive and mitigate the potential adverse effects of the obstetric intervention [24]."
The study does not prove it. Should be clarified, as it may imply causality not demonstrated by the study.
Response 15: Thank you for your comment. We agree with your comment. The sentence has been rewritten in page 11, lines 291-294.
Point 16: The statement: "This suggests that multiparity does not always guarantee higher EBF rates, possibly influenced by prior unsatisfactory experiences."
can be expanded to mention other influencing factors.
Response 16: Thank you for your comment. This point and next has been argued in response 17.
Point 17: The claim: "These findings are consistent with some previous studies [26,27]."
should be balanced by citing studies with conflicting results.
Response 17: Thank you for your comment. Your suggestion has been incorporated into the text. A reference has been added and the sentence you pointed out has been completed, in page 11, lines 304-305 and a new reference has been added.
Point 18: Regarding breech presentation and cesarean delivery: "Breech presentations that result in caesarean births may delay the initiation of breastfeeding and impair the mother’s ability to breastfeed [34–36]."
How does breech presentation delay the start of breastfeeding? It would be helpful to specify how breech presentation delays breastfeeding initiation.
Response 18: Thank you for your comment. Breech presentation delays breastfeeding through the associated caesarean section, as the latter often leads to a delay in skin-to-skin contact and the first feed, increased post-surgical pain that hinders positioning, and the possible influence of anaesthetic/analgesic medication on newborn alertness. We have added a new paragraph in page 12, lines 324-327.
Point 19: The sentence: "Although higher Apgar scores facilitated effective breastfeeding at 36 hours, this study did not find a clear association with long-term success [44]." Which study? If it is yours, the breastfeeding data have been analyzed until discharge and not in the long term, so this sentence seems improper. Should specify which study is referenced. If your data only cover discharge, long-term conclusions are limited.
Response 19: Thank you for your comment and your precise observation. We confirm that our study only analysed breastfeeding success up to hospital discharge.
The phrase regarding the lack of a clear association with 'long-term success' referred directly to the study cited in reference [44], not to a conclusion derived from our own data.
We have modified the sentence to clearly specify that the lack of long-term association in page 12, lines 364-365.
Point 20: Speaking of this, among the limitations in my opinion is the fact that the data are related to discharge and therefore, as interesting as they are, they are a bit limiting because I imagine that you too discharge early at 48-72 hours maximum and it should be noted that the most significant data would be that relating to the following months. The limitations related to data collection at discharge—often within 48-72 hours—and the absence of follow-up data over months should be explicitly acknowledged.
Response 20: Thank you for your input. We agree with your reflection, and we have included a new sentence in the limitations section as follows: “A crucial limitation is that EBF was assessed only at hospital discharge (48–72 hours), an early cutoff point that is susceptible to bias and prevents analysis of its persistence across Robson groups long-term.” Page 14, lines 455-457.
Point 21: Also, discuss how these limitations influence interpretation and future research.
Response 21: Thank you for your comment. We have included this sentence as follows: However, given that this is the first study to address the relationship between EBF and the RTGCS, it establishes a fundamental basis, and future studies should be conducted, especially with prospective follow-up up to 6 months, to contrast the persistence of these results. Page 14, lines 458-461.
Conclusions
Point 22: Conclusions should be more concise and impactful, clearly summarizing the key messages without repetition.
Response 22: Thank you for your comment. We accept your suggestion. We have condensed and focused the Conclusions section to emphasize the role of the RTGCS and our key findings in a more concise manner, as you requested, in page 15, lines 491-497.
Round 2
Reviewer 1 Report
Comments and Suggestions for Authors
The author revised the questions I put forward and improved the quality of the manuscript.
Author Response
Comment: The author revised the questions I put forward and improved the quality of the manuscript.
Response: Thank you very much for your valuable feedback throughout the this submission process.
Reviewer 3 Report
Comments and Suggestions for Authors
The text is much improved and can be published. Two small clarifications.
In our study, 74.783% of women who had a vaginal 313 delivery practiced EBF, a percentage that dropped to 71.955% among those who 314 underwent a caesarean section. Moreover, previous studies have reported that surgical 315 interventions and the administration of medications during caesarean delivery can 316 adversely affect the maternal ability to breastfeed during the early postpartum period 317 [33,34]. Thus, the low EBF rates observed in these groups likely reflect the high prevalence 318 of surgical interventions.
It would be useful to specify the percentage of reduction which is not dramatic and is not consistent with the following comment that must be mitigated.
Stillbirth was more frequent in the preterm Group 10, consistent with 366 recent literature [48].
It seems obvious to me that prematurity is associated with higher mortality. I wouldn't put it
Author Response
Response to Reviewer 3 Comments_Round2
Thank you for your very positive and constructive feedback on our manuscript.
We have considered all your comments and suggestions (and the comments made by the other reviewer), attempting to improve/refine the original manuscript.
Below you will find a point-by-point response to your comments (in red).
The text is much improved and can be published. Two small clarifications.
Point 1: In our study, 74.783% of women who had a vaginal delivery practiced EBF, a percentage that dropped to 71.955% among those who underwent a caesarean section. Moreover, previous studies have reported that surgical interventions and the administration of medications during caesarean delivery can adversely affect the maternal ability to breastfeed during the early postpartum period [33,34]. Thus, the low EBF rates observed in these groups likely reflect the high prevalence of surgical interventions.
It would be useful to specify the percentage of reduction which is not dramatic and is not consistent with the following comment that must be mitigated.
Response 1: We appreciate the constructive comment and the opportunity to clarify the magnitude of the difference observed in Exclusive Breastfeeding (EBF) rates at discharge. As recommended by the Editor, we have adjusted the wording in the Discussion. We have removed the mention of the absolute percentage difference from the Discussion text to avoid ambiguity and focus on the clinical interpretation. Our argument now focuses exclusively on the Robson Group analysis, which provides a much more nuanced and significant view of how the specific surgical intervention is associated with a significantly higher risk of EBF failure.
Point 2: Stillbirth was more frequent in the preterm Group 10, consistent with recent literature [48].
It seems obvious to me that prematurity is associated with higher mortality. I wouldn't put it
Response 2: We thank the reviewer for their observation. We have rewritten this sentence in page 11 lines 373-376.